# Tea Tree Oil Mediates Antioxidant Factors Relish and Nrf2-Autophagy Axis Regulating the Lipid Metabolism of *Macrobrachium rosenbergii*

**DOI:** 10.3390/antiox11112260

**Published:** 2022-11-16

**Authors:** Mingyang Liu, Xiaochuan Zheng, Cunxin Sun, Qunlan Zhou, Bo Liu, Pao Xu

**Affiliations:** 1Wuxi Fisheries College, Nanjing Agricultural University, Wuxi 214081, China; 2Key Laboratory of Aquatic Animal Nutrition and Health, Freshwater Fisheries Research Center, Chinese Academy of Fishery Science, Wuxi 214081, China

**Keywords:** tea tree oil, lipid metabolism, autophagy, Relish, Nrf2, *Macrobrachium rosenbergii*

## Abstract

Both oxidative stress and autophagy refer to regulating fat metabolism, and the former affects autophagy, but the role and mechanism of the antioxidant–autophagy axis in regulating lipid metabolism remains unclear. As an antioxidant, tea tree oil (TTO) has little research on the regulatory mechanism of lipid metabolism in crustaceans. This study investigated whether TTO could alter hepatopancreatic lipid metabolism by affecting the antioxidant–autophagy axis. Feed *Macrobrachium rosenbergii* with three different levels of TTO diets for 8 weeks: CT (0 mg/kg TTO), 100TTO (100 mg/kg TTO), and 1000TTO (1000 mg/kg TTO). The results showed that 100TTO treatment reduced the hemolymph lipids level and hepatopancreatic lipid deposition compared to CT. In contrast, 1000TTO treatment increased hepatopancreatic lipid deposition, damaging both morphology and function in the hepatopancreas. The 100TTO treatment promoted lipolysis and reduced liposynthesis at the transcriptional level compared to the CT group. Meanwhile, it improved the hepatopancreas antioxidant capacity and maintained mitochondrial structural and ROS homeostasis. In addition, it simultaneously activated the expression of transcription factors *Keap1*-*Nrf2* and *Imd*-*Relish*. By contrast, the 1000TTO group significantly enhanced the ROS level, which considerably activated the *Keap1*-*Nrf2* signaling expression but had no significant effects on the expression of *Imd*-*Relish*. The 100TTO group supplementation significantly enhanced lipid droplet breakdown and autophagy-related genes and protein expression. On the contrary, the 1000TTO group significantly inhibited the expression of genes and proteins related to autophagy. Pearson analysis revealed that *Nrf2* has a positive correlation to lipid anabolism-related genes (*Fasn*, *Srebp1*, *Pparγ*) and autophagy regulators (*mtor*, *akt*, *p62*), and were negatively correlated with lipolysis-related genes (*Cpt1*, *Hsl*, *Ampkα*) and autophagy markers (*Ulk1*, *Lc3*). *Relish* was positively correlated with *Atgl*, *Cpt1*, *Ampkα*, *Ulk1*, and *Lc3*, and negatively correlated with *Pparγ* and *p62*. Moreover, *Keap1* and *Imd* were negatively correlated with *p62* and *mtor*, respectively. In sum, 100 mg/kg TTO enhanced antioxidant activity and increased autophagy intensity through the *Relish*-*Imd* pathway to enhance lipid droplet breakdown, while 1000 mg/kg TTO overexpressed *Nrf2*, thus inhibiting autophagy and ultimately causing excessive lipid deposition and peroxidation. Our study gives a fresh perspective for deciphering the bidirectional regulation mechanism of lipid metabolism by different doses of TTO based on the antioxidant–autophagy axis.

## 1. Introduction

For quite some time, the ongoing evolution of feed standards has gradually enhanced the nutritional quality of aquatic feed and expanded the economic benefits of aquaculture [1]. However, high-frequency feeding and excessive nutrient intake (high fat and carbohydrate content in aquafeeds) derived from the pursuit of benefits lead to eutrophic diseases, such as lipid metabolism disorder [2].

The hepatopancreas is the crucial tissue for lipid anabolism and deposition to the crustacean. The hepatopancreas of crustaceans has similar functions to the pancreas, liver, and intestine in mammals [3], and it is more sensitive and intolerant to high lipid deposition than the liver of mammals [4]. The excessive deposition of lipids can lead to steatosis-like lesions of the hepatopancreas [5]. The pathological symptoms of lipid metabolism disorders in crustaceans are similar to that of mammalian nonalcoholic fatty liver disease, manifested by abnormal blood lipid parameters and the disordered expression of hepatic lipid synthesis and catabolism genes (*Acca*, *Fasn*, *Pparα*, *Cpt1*) [6,7].

Previous studies pointed out that lipid accumulation and metabolism were regulated by oxidative stress [8,9]. Abnormal lipid metabolism induced by oxidative stress is an essential reason for the occurrence of lipid metabolic diseases [10,11]. Notably, autophagy was found to be an important intermediate mechanism of oxidative stress involved in the regulation of lipid metabolism [12]. Normal cellular processes require appropriate ROS levels, but excessive ROS can induce oxidative stress and cause severe damage to cell structure and function [13,14]. Simultaneously, ROS can affect the essential regulatory pathway of autophagy Ampk/mTOR to induce autophagy [15]. Oxidative stress and ROS levels are not only regulated tightly by the Keap1 and Nrf2 proteins [16,17] but are also affected by the *NF-κB* immunity regulatory pathway [18]. Further, new reports found that these antioxidant genes combined with the Ampk-mTOR pathway could form a discontinuous adverse feedback pathway to affect autophagy [19,20].

Tea tree oil (TTO) is a typical essential oil of flora, also known as *Melaleuca alternatifolia* essential oil, which is obtained from distilled *Melaleuca alternatifolia* leaves [21], well known for its antioxidant and growth-promoting functions [22]. Our earlier study exhibited that plant essential oils could activate *NF-κB* and regulate oxidative stress [23]. Essential oils can regulate blood lipid levels and lipid peroxidation [24]. Additionally, study found that an excessive increase in ROS is a critical factor in lipid excess in the liver [25]. High levels of essential oils induce oxidative stress and increase LPO and ROS production [26,27]. Active ingredients of *Calocedrus formosana* essential oil have also been found to activate lipid autophagy by significantly increasing *Ampk* phosphorylation [28]. The impairment of lipid autophagy is direct to liver metabolic diseases, and the gene level is shown as the reduction in vital autophagic genes such as *atg5* or *atg7* [29]. Therefore, the up-regulation of autophagy is considered beneficial, and autophagy impairment is a phenotype of several liver diseases [30,31]. Few studies have been conducted based on the “antioxidant-autophagy-lipid metabolism” axis to explore the regulation mechanisms of lipid metabolism by tea tree oil, let alone explicitly targeting these two oxidative factor regulators (*NF-κB* and *Nrf2*).

*Macrobrachium rosenbergii* is a frequently used crustacean model animal for immunology, nutrition, and metabolism research. This study used *M. rosenbergii* as a model animal to explore whether different TTO levels regulate lipid metabolism through the “antioxidant-autophagy-lipid metabolism” axis and its key regulatory targets. It is essential for the development of therapeutic strategies for oxidative stress-type lipid metabolic diseases.

## 2. Materials and Methods

### 2.1. Ethical Statement

All experiments were conducted following the guidelines for scientific breeding and use of animals formulated by the IACUC under the Chinese Academy of Fishery Sciences. At the same time, it has followed the guidelines of the Institute of Animal Research of Nanjing Agricultural University on animal care and use.

### 2.2. Experimental Materials, Design, and Growth Evaluation

Dietary formulation and *M. rosenbergii* feeding were presented in Table 1. We formulated experimental diets containing three different levels of TTO at 0 ppm (diet without TTO, CT), 100 ppm (diet with 100 mg/kg TTO, 100 TTO), and 1000 ppm (diet with 1000 mg/kg TTO, 1000 TTO) (Table 1). Put 360 freshwater prawns (initial body weight: 0.15 ± 0.01 g) in 9 vats (R × H, 1.0 m × 1.5 m), randomly assigned to 3 groups with 3 replicates, each with 40 prawns. During the domestication period, the prawns were fed with commercial feed provided by Fuyuda Food Products Co. LTD., Yangzhou, China for one week. The prawns were fed a day thrice at 8:00 am, 1:00 pm, and 6:00 pm. After 8 weeks of feeding, the survival rate and the weight of prawns in each group were recorded. Calculation methods of feed the conversion ratio (FCR), hepatosomatic index (HSI), specific growth rate (SGR), and weight gain rate (WGR) were listed as follows:WGR (%) = (Final weight − Initial weight)/initial weight × 100SGR(%/day) = (Ln final weight − Ln initial weight) × 100/daysFCR = Dry feed intake (g)/weight gain (g)HSI (%) = (Hepatopancreas weight/ final weight) × 100

### 2.3. Samples Collection

After 8 weeks of feeding, in order to avoid the influence of diet, the prawns were fasted for 12 h and then euthanized with MS-222 (Millipore Sigma, St. Louis, MO, USA). The hepatopancreases of 3 prawns were dissected from each vat for samples. The samples were put into cryogenic vials, thrown into liquid nitrogen for quick freezing, and finally, stored in a refrigerator at −80 °C for subsequent RNA and protein separation. In order to analyze the biochemical indexes of hemolymph, 3 prawns were randomly picked from each vat for sampling, the hemolymph and blood cells were separated at 4 °C for 4000 rpm and 10 min, and the supernatant was put into a 4 °C refrigerator with a 1.5 mL centrifuge tube [32]. In addition, the hepatopancreases of 3 prawns were taken for immunohistochemical and ultrastructural analysis and put into 4% paraformaldehyde solution and 2.5% glutaraldehyde solution (Biosharp, Labgic, Technology Co., LTD., Hefei, China) with 5 mL centrifuge tubes, respectively. The remaining samples were stored in the refrigerator at −80 °C for standby.

### 2.4. Sample Analysis

#### 2.4.1. Antioxidant, Peroxide, and Lipid Parameters Analysis

Catalase (CAT), glutathione peroxidase (GSH-Px), total antioxidant capacity (T-AOC), total superoxide dismutase (T-SOD) activities, protein carbonyl (OH), and triglyceride (TG) content used Nanjing Jiancheng Institute of Bioengineering kits for measuring. The general measurement method is to suspend the hepatopancreas in 50 mM potassium phosphate buffer, which contains 0.5 mM EDTA, and the pH value is about 7.0. Use a handheld homogenizer to homogenize the hepatopancreas for 5 min in a 2.0 mL centrifuge tube. After homogenization, all samples were centrifuged at 4 °C for 15 min at 2000× *g*. The supernatant product was extracted for enzyme analysis, and the Spectra Max Plus spectrophotometer (Molecular Devices, Menlo Park, CA, USA) was used for full-package enzyme activity analysis. The content of malondialdehyde (MDA) in the hepatopancreas was determined by the thiobarbituric acid (TBA) test [33]. Nanjing Jiancheng Bioengineering Research Institute provided the determination kit. The malondialdehyde extraction of the supernatant for determination is similar to the enzyme method. All specific experimental methods are according to the manufacturer’s instructions.

According to the manufacturer’s instructions, the content of several lipid metabolism-related enzymes (Acetyl-CoA carboxylase, ACC; carnitine palmitoyltransferase 1, CPT1; fatty acid synthase, FAS) and peroxide (4-hydroxynonenal, 4-HNE; Lipid peroxidation, LPO) was determined by ELISA kits (Shanghai mlbio Biotechnology Co., Ltd., China) specific for freshwater shrimp. Briefly, the hepatopancreas supernatant was mixed with the reactants and recorded the absorbance corrected at 430 nm and 650 nm. The Spectra Max Plus spectrophotometer was used for all of Elisa’s technical readings of optical density.

#### 2.4.2. ROS Levels

Flow cytometry was used based on a previous study that analyzed ROS levels [34] in the fresh hemolymph cell of *M. rosenbergii*. Cells were washed three times by PBS (1000 rpm for 5 min), and the final concentration of the cell suspension was adjusted to 1 × 10^6^/mL. Then, use the serum-free medium to dilute DCFH-DA to a concentration of 10 m at 1:1000, collect cells, suspend them in diluted DCFH-DA, and incubate them at 37 °C for 20 min. Then, mix them upside down for 3–5 min to let the probe and the cells fully contact. Next, the cells were washed three times using serum-free cell culture to remove DCFH-DA that does not fully enter cells. Finally, flow cytometry was used to detect (Ex = 488 nm; Em = 530 nm) ROS in cells.

#### 2.4.3. Oil Red O Staining, H&E Stainings, and Transmission Electron Microscopy

Remove fixed hepatopancreas samples in 4% paraformaldehyde buffer. The samples were embedded in the optimum cutting temperature (OCT) medium and stored at −80 °C. For OCT embedding, 7 μ of the fat content of the m sections was detected by ORO (Sigma Aldrich, St. Louis, MO, USA) staining. Use and equip an Olympus UPlan Apo 200×. The objective lens of the Olympus BX51 optical microscope (Tokyo, Japan) was used to take pictures of randomly selected tissue sections. ImageJ software (National Institutes of Health, Bethesda, MD, USA) adapted from the ORO-staining section algorithm was used to analyze the lipid content of the ORO-staining section and expressed it as the % of the area of lipid-positive hepatopancreas tissue and finally determined its gray value. According to published methods, Hematoxylin-eosin (H&E) and transmission electron microscopy staining for hepatopancreas were conducted [32,35]. To calculate the relative area of the liver vesicles in the H&E stainings, 3 fields were analyzed randomly. Additionally, then, Image J software was used for quantification.

#### 2.4.4. Haemolymph Biochemical Parameters

The levels of hemolymph alanine aminotransferase (ALT), aspartate aminotransferase (AST), albumin (ALB), and lactate dehydrogenase (LDH) were measured by an automatic hemolymph biochemical analyzer (Mindray BS-400, Shenzhen, China). The specific method is to put 200 μL hemolymph into a special colorimetric tube according to the manufacturer’s instructions and use the corresponding commercial kit (Mindray Medical International Co., Ltd., Shenzhen, China).

### 2.5. Real-Time Quantitative PCR (qPCR)

According to the fragments obtained from the original hepatopancreas transcriptome determined initially in our laboratory and the ORF intercepted, primer 5.0 was used to design the gene primer for qRT PCR analysis (see Appendix A). Because the expression of *β-actin* is stable [23], choosing *β-actin* mRNA serves as an internal reference. Shanghai Sangong Biotechnology Co., Ltd. synthesized PCR primers.

The total RNA of the hepatopancreas in the three experimental groups (3 samples for each repetition) was extracted with RNAiso Plus (TaKaRa, Japan). Then, the concentration was determined with Nanodrop 2000 (Thermo Fisher Scientific, Waltham, Massachusetts, USA). The RNA concentration of each sample was diluted to 500 ng/mL. Use Two-Step SYBR^®^ Prime Script^®^ Plus RT-PCR Kit (TaKaRa, Japan) for 2 μg quantitative analysis of total RNA; the total system of sample loading was 20 μL. Based on Liu et al. [36], an ABI 7500 Real-time PCR system was used for real-time quantitative PCR (RT-PCR), and the 2^−ΔΔCT^ method was used to calculate the relative gene expression.

### 2.6. Western Blot Analysis

We performed Western blot analysis with reference to Dai et al. [37]. Simply put, the protein was extracted from hepatopancreas tissue with a high-fat protein extraction kit (BB3162, Bestbio Biotechnology Co., LTD., Shanghai, China) and we performed Western blot by the manufacturer’s guidelines (Bio/Rad, Hercules, CA, USA). The final strip was used for imaging. Enhanced chemiluminescent substrate reagents (CW0049-S, Cwbio Biotechnology Co., LTD, Shanghai, China) were used to detect immune complexes, visualization by a luminous image analyzer (ChemiDoc XRS+, Bio/Rad, USA) according to the manufacturer’s instructions. In the experiment, the GAPDH antibody (1:10,000, Proteintech, Chicago, IL, USA, 66009-1-Ig), Relish/NF-kB (1:1000, Abcam, ab183559), Nrf2 (1:2500, Proteintech, 16396-1-AP), Beclin 1 antibody (1:5000, Proteintech, 11306-1-AP), Ulk 1 antibody (1:4000, Cell Signaling Technology, Danvers, MA, USA, #8054S), and Lc3 antibody (1:2000, Abcam, ab128025) were used. Image J software (NIH, Bethesda, MD, USA) was used for gray-scale analysis to evaluate protein band intensities.

### 2.7. Data Analysis

After Levene tested the homogeneity of variance, the data were subjected to a two-way ANOVA analysis to investigate the evaluation of growth, enzyme activity, hemolymph index, and relative gene and protein expression. Suppose the difference is significant (*p* < 0.05), the Duncan multi-interval test is used to rank the mean. Pearson correlation analysis was used to analyze the correlation between the two variables that conform to the normality and used the following marks: one asterisk (*) indicated a significant difference (*p* < 0.05), and two asterisks (**) indicated an extremely significant difference (*p* < 0.01). All data were demonstrated as means ± S.E.M. (standard error of mean). All the above analysis methods use SPSS program v16.0 (SPSS Inc., Michigan Avenue, Chicago, IL, USA) on Windows 10.

## 3. Results

### 3.1. Growth Evaluation

Table 2 showed that the survival rate (SR) was lower for prawns fed the 1000 mg/kg TTO diet (*p* < 0.05). The 1000TTO group had a higher weight gain rate (WGR) and significantly increased specific growth rate (SGR) and hepatosomatic index (HSI) than the other two groups (*p* < 0.05). The WGR and SGR in the 100TTO group were higher than the control but had no significant difference (*p* > 0.05). The feed conversion ratio (FCR) and HSI in the 100TTO group were lower than in the CT and 1000TTO group (*p* < 0.05).

### 3.2. Morphological Characteristics of Hepatopancreas

Figure 1 showed that the dietary TTO levels considerably affected the hepatopancreatic morphology and lipid deposition. Prawns fed 100 mg/kg TTO diet significantly decreased the vacuolization degree in hepatopancreas than the CT (*p* < 0.05), and the 1000TTO group significantly increased the vacuolization degree (*p* < 0.05, Figure 1B). Compared with the CT and 1000TTO groups, the hepatopancreatic nuclei in the 100TTO group were more visible, and the cell boundary was more pronounced. The hepatopancreatic lipid droplet area of the 100TTO group was significantly decreased compared to the other two groups (*p* < 0.05, Figure 1C). The highest lipid droplet area was found in the 1000TTO group but without significant difference from the control (*p* > 0.05). Compared with the CT and 100TTO groups, the hepatopancreatic cells in the 1000TTO group were significantly swollen.

### 3.3. Hepatopancreatic Cell Function

As shown in Figure 2, dietary TTO levels significantly affected hepatopancreatic cell function. Compared to the control, 1000TTO induced significantly increased activity levels of hepatopancreatic function-related enzymes (AST, ALT, and LDH) and a significantly decreased ALB level (*p* < 0.05). There were no significant differences in the AST, ALT, ALB, and LDH levels between the 100TTO group and the control group (*p* > 0.05).

### 3.4. Lipid Contents in Hepatopancreas

Compared to the CT group, TG and LDL-C significantly decreased in the 100TTO group, while the TG content in the 1000TTO group significantly increased (*p* < 0.05, Figure 3A–C). In addition, although the TC content in the 100TTO group was lower than that in the control group, it had no consequential difference (*p* > 0.05). The 1000TTO group significantly increased TC and LDL-C compared to the 100TTO group (*p* < 0.05). At the same time, the 1000TTO group had a lower HDL-C level than the other two groups (*p* < 0.05, Figure 3D).

### 3.5. Lipid Metabolism Enzymes Activities and Gene Expression

Dietary TTO levels significantly affected lipid metabolism enzymes in the hepatopancreas (Figure 4A–C). Compared to the CT group, the 100TTO group had significantly lower FAS and ACC levels and higher CPT1 levels (*p* < 0.05). The 1000TTO group had higher FAS and ACC contents than the control. CPT1 content showed no significant differences between the 1000TTO group and the control (*p* > 0.05).

At the transcriptional level (Figure 4D), the 100TTO group had significantly lower mRNA abundances of lipogenic genes (*6pgd*, *Acca*, *Fasn*, and *G6pd*) and *Pparγ*, and higher lipolytic (*Cpt1*, *AtgL*, *Hsl*, *Ampkα*) and *Pparα* mRNA expression compared to the control (*p* < 0.05). At the same time, the 1000TTO group significantly increased the 6pgd, *Fasn*, *Srebp1*, *Fabp7*, and *Pparγ* mRNA expression, and decreased lipolytic gene expression (*Cpt1* and *Ampkα*) compared to the control (*p* < 0.05). *Dagt* and *Fatp1* mRNA abundances showed no significant differences among the three groups (*p* > 0.05).

### 3.6. Antioxidant Enzyme Activity and Peroxidase Levels in Hepatopancreas

The dietary TTO levels significantly changed the activities of antioxidant enzymes in the hepatopancreas (Figure 5). Compared to the CT group, the levels of CAT, T-SOD, GSH-PX, and T-AOC in the 100TTO group significantly increased, the content of MDA significantly decreased (*p* < 0.05), and the PCO activity had no significant difference between the two groups (*p* > 0.05). In the 1000TTO group, considerably increased MDA and PCO activities were found compared to the control (*p* < 0.05). Additionally, the levels of T-SOD in the 1000TTO group were significantly lower than in the control, and the CAT and T-AOC activities were lower than those in the control but had no significant distinction (*p* < 0.05).

### 3.7. Different TTO Levels Affect Oxidative Stress in the Hepatopancreas

Ultrastructural changes in mitochondria were given in Figure 6A. In the CT and 100TTO group, mitochondria showed standard morphology under TEM. The double membrane and mitochondrial ridge of the nucleus were excellently visible. However, the mitochondria damage was found in the 1000TTO group, accompanied by the mitochondria turgidity, vacuolation, and the ridge broken. At the same time, the nuclear membrane of the 1000TTO group was blurred, and pyknosis deformity occurred.

The 1000TTO group significantly increased ROS levels more than the other two groups (Figure 6B). The ROS level of 100TTO increased slightly compared with the control group. At the same time, ELISA results showed that the 1000TTO increased the LPO and 4-HNE contents compared to the control and 100TTO groups (*p* < 0.05, Figure 6C,D). The LPO content of the 100TTO group was extremely lower than the control (*p* < 0.05).

mRNA expressions of significantly lower *Keap1* and higher *Nrf2*, *Imd*, and *Relish* were found in the 100TTO group compared to the CT (*p* < 0.05, Figure 6E). In addition, the 1000TTO group substantially decreased *Keap1* and increased *Nrf2* mRNA expression compared to the control (*p* < 0.05). It is worth noting that the 1000TTO group significantly decreased *Imd* and *Relish* mRNA expression more than the 100TTO group (*p* < 0.05). Figure 6F,G showed that the 100TTO group had higher Relish protein expression than the 1000TTO group and higher Nrf2 than the control (*p* < 0.05). The 1000TTO group had a higher Nrf2 protein expression diet than the other two groups.

### 3.8. Autophagy-Related Genes and EM Observation

The prominent display of the electron microscope (EM) pictures of the prawn hepatopancreas in each group can see the difference in the degree of autophagy (Figure 7A). The 100TTO group promoted the formation of autolysosomes and autophagosomes and hardly found large lipid droplets, while a small amount of autolysosome formation around some large lipid droplets was found in the control. Numerous large lipid droplets and thimbleful autophagosomes were found in the 1000TTO group. The quantification of autophagic vesicles by TEM confirmed that autophagic changes were in response to the level of TTO. To further analyze the effect of the TTO level on autophagy, we measured the autophagy-related genes.

As shown in Figure 7B, the 100TTO group significantly regulated the expression of autophagosome upstream regulatory genes (*Ampk* increased, *mtor* decreased) and autophagy markers (*Lc3* increased and *p62* decreased) compared to the control (*p* < 0.05). At the same time, the 100TTO group significantly up-regulated the expression of autophagosome membrane initiation (*Ulk1* and *Beclin1* increased) and autophagosome membrane expansion-related genes (*atg13*, *atg3*, *atg4b*, *atg5*, and *atg7* increased). The 1000TTO group significantly down-regulated *Ampk*, *Beclin1*, *atg13*, *atg5*, and *Lc3* expression and up-regulated *akt* and *mtor* expression compared to the control (*p* < 0.05). Meanwhile, the 1000TTO group significantly down-regulated the autophagy marker gene *Lc3* accompanied by significantly up-regulated autophagic flux gene *p62* (*p* < 0.05). Western blotting analysis showed that prawns fed the 100 mg/kg TTO diet had higher Beclin1, Ulk1, and Lc3 II protein expression than those in the control and the 1000TTO group (*p* < 0.05, Figure 7C,D). Additionally, the 1000TTO group had lower Beclin1 protein expression than the control group (*p* < 0.05).

### 3.9. Correlation Analysis

Figure 8 showed the Pearson correlation analysis of oxidative stress regulatory gene mRNA levels with lipid anabolism and autophagy genes. We found that the mRNA levels of *Fasn*, *Srebp1*, and *Pparγ* were positively correlated with *Nrf2*. The levels of *Cpt1*, *Hsl*, and *Ampkα* were negatively correlated with *Nrf2*. In addition, the mRNA expression of *Atgl*, *Cpt1*, and *Ampkα* were positively correlated with *Relish*, and *Pparγ* was negatively correlated with *Relish*.

Comparing oxidative stress and autophagy (Figure 8B), the *Nrf2* levels were positively correlated with *mtor*, *akt*, and *p62* and negatively correlated with *Beclin1* and *Lc3*. The *Relish* levels were positively correlated with *Ulk1* and *Lc3* and negatively correlated with *p62*. Meanwhile, *Keap1* and *Imd* were negatively correlated with *p62* and mtor, respectively.

## 4. Discussion

In the present study, we found that different TTO levels significantly affected the growth performance of *M. rosenbergii*. The prawns fed with a 100 mg/kg TTO diet improved the growth indicators (WGR, SGR, FCR) while decreasing the HSI compared to the control. Decreased HSI may be associated with enhanced hepatic lipid energy metabolism and the energy used for the growth of other tissues [38]. Therefore, combined with our undisclosed results about muscles, we speculated that the 100 mg/kg TTO diet may promote the other tissues’ growth (e.g., muscles) instead of the hepatopancreas. An important finding in our study is that the 1000TTO group significantly improved growth indicators (WGR, SGR) with a substantial increase in HSI compared to the control. This result indicated that the 1000TTO group may improve growth performance mainly by increasing hepatopancreas weight. However, the marked increase in HSI may be due to burgeoning lipid accumulation, often leading to metabolic disease [39]. At the same time, in crustaceans, the HSI surge also damages the tissue structure of the hepatopancreas [40], which verified that the survival rate of the 1000TTO group was significantly decreased compared to the other two groups (CT and 100TTO). The hepatopancreas is a vital metabolic organ of crustaceans, and its main component is lipids. Furthermore, excessive lipid deposition in the hepatopancreas not only leads to damage to the hepatopancreatic metabolic function and eventually leads to the lesions of the hepatopancreas but also impairs the antioxidant capacity [41,42]. Based on this premise, we speculate that 1000 mg/kg TTO causes a decreased survival rate and brings about lesions caused by superfluous lipid production in the hepatopancreas. However, 100 mg/kg TTO could reverse this phenomenon. Based on this speculation, we further measured the lipid deposition status in the hepatopancreas.

In the ORO staining results, we found significantly increased lipid droplet content and steatosis in the hepatopancreas in the 1000TTO group. Furthermore, we found the vacuolization of hepatopancreatic cells induced by 1000TTO supplementation through H&E analysis. Moreover, the nuclei of hepatopancreatic cells and the cell boundary were not obvious in the 1000TTO group. This phenotype was a classical manifestation of hepatopancreatic injury in previous studies [43]. In contrast, the 100TTO group had significantly lower lipid droplet content, less vacuolation, and a clearer nucleus. This proves the conclusion that the 1000TTO group will cause excessive fat accumulation and hepatopancreas tissue damage, while 100TTO treatment maintains lipid homeostasis and protects hepatopancreatic cells.

In addition, when there is excess fat deposition in the hepatopancreas and the organism hardly metabolizes the excess fat, it can lead to elevated blood lipids [44]. The present study found that the 100TTO group down-regulated blood lipid indicators while the 1000TTO group increased blood lipid contents. Therefore, we speculated that the dysfunction of the hepatopancreas in the 1000TTO group resulted in the inability to metabolize excess lipids. To assess its health status, we measured indicators reflecting hepatopancreatic function (AST, ALT, ALB, LDH). Compared with the control, 100TTO treatment had no adverse effect on hepatopancreatic function. However, the 1000TTO treatment may destroy hepatopancreatic function supported by up-regulated AST, ALT, and LDH levels and down-regulated ALB content in the hemolymph. The significantly elevated AST, ALT, and LDH in the hemolymph are markers of impaired hepatopancreatic function. When the hepatocytes are damaged, AST and ALT will enter the blood from the hepatocytes and cause their content to increase [45,46]. ALB is mainly synthesized in the hepatopancreas and is an important immune protein. Hepatopancreatic function damage will bring about a significant reduction in ALB levels [47]. Combined with AST, ALT, LDH, and ALB results, it was proved that 1000TTO causes hepatopancreas dysfunction. In summary, 100TTO treatment did reduce the fat content in the hepatopancreas and did not cause damage to the hepatopancreas. In contrast, the 1000TTO group increased fat deposition and disrupted hepatopancreas morphology and function.

In order to elucidate the mechanism of TTO influencing lipid deposition and metabolism, we further explored changes in lipid metabolism-related enzymes and genes in the hepatopancreas. Our ELISA results showed that the essential enzymes of fat synthesis (FAS and ACC) [48] observably decreased in the 100TTO group and increased in the 1000TTO group. In addition, CPT1 is a rate-limiting enzyme in fatty acid β-oxidation [49]; the significant increase in CPT1 in the 100TTO group indicated that the 100TTO treatment stimulated fatty acid β-oxidation and accelerated lipid breakdown. At the transcriptional level, we found that 100TTO up-regulated the expression of lipolysis-related genes and down-regulated the expression of fat synthesis-related genes in the hepatopancreas. The trends in the 1000TTO group are the opposite of the 100TTO group.

In terms of the transcription factor, our study found that the 1000TTO group increased the mRNA abundances of *Srebp1*, *Fatp1*, and *Pparγ*, which also confirmed its promotion of fat synthesis. Existing evidence shows that *Fatp1* is a crucial transcription factor for regulating lipid synthesis, responsible for fatty acid transport [50]. *Srebp1* and *Pparγ* are classical lipogenic regulators [51]. *Pparα* is the transcription factor regulating lipid metabolism and could activate *Cpt1* to improve steatosis induced by excess lipid deposition [52]. In this study, only the 100TTO group enhanced the *Pparα* levels. Our data suggest that the 100TTO group decreased lipid synthesis and promoted lipolysis and fatty acid β-oxidation. In contrast, the 1000TTO group promoted lipid production while inhibiting the breakdown of deposited lipids.

Autophagy is a physiological process of self-repair of cells and subcellular structures after being subjected to stress, injury, or free radicals and plays an important role in maintaining lipid metabolism homeostasis [53]. The inhibition of autophagy has been observed in some models of dyslipidemia or disorders [54]. At the same time, the principle of some lipid metabolism-modulating drugs and additives to regulate excessive lipid deposition is to regulate autophagy [55,56]. In general, autophagy plays many important roles in regulating hepatocyte lipid metabolism. Lipid droplets can serve as substrates for the process of autophagy, so autophagy plays a role in regulating the lipolysis process [57]. Because of the vital role of autophagy in lipid metabolism, we further explored the effect of TTO on autophagy.

Through ultramicroscopic analysis, we hardly observed the phenotype of autophagy in the 1000TTO group, and the lipid droplet morphology was intact compared with a small number of autophagolysosomes existing in lipid droplets in the CT group. In sharp contrast, the 100TTO group had a large number of autophagosomes in lipid droplets accompanied by the breakdown of lipid droplets. These results showed that the 100TTO group enhanced autophagy, possibly decomposing lipids through autophagy, while the 1000TTO group inhibited autophagy and caused lipid deposition in the hepatopancreas. Degraded lipids by enhanced mitophagy and lipophagy were commonly found in other studies [58,59], but there was seldom research on essential oils affecting lipid metabolism by regulating autophagy.

Autophagy is a process by which a membrane derived from the organelle or lysosome surrounds the material for degradation to form an autophagosome [30]. At the molecular level, the mRNA results showed that 100 mg/kg TTO activates autophagy by initiating autophagic membrane phagocytosis and regulating autophagy-related gene expression. The 100 mg/kg TTO enhanced autophagosome membrane initiation and the expansion of mRNA expression, and 1000 mg/kg TTO inhibited the generation and expansion of autophagosome membrane mRNA expression. Meanwhile, the WB analysis showed that 100TTO intervention up-regulated the expression of two autophagy marker proteins (*Beclin1* and *Lc3*) and increased the expression level of autophagy initiation protein *Ulk1* and the degrading cell protein *atg7* [60]. In addition, we found that 100TTO increased the gene expression of the upstream activator proteins *Ulk1* and *Ampk*, and the 1000TTO group increased the expression of the upstream autophagy-inhibiting protein *mTOR* [61]. Moreover, a higher expression of *p62* was found in the 1000TTO group, indicating that the autophagy was suppressed, as the high intracellular expression of *p62* occurs when autophagy is blocked [62]. The above results indicated that the 100TTO treatment activated the autophagy upstream essential proteins to enhance the formation of the autophagy membrane. In contrast, the 1000TTO group down-regulated the expression of autophagy-related proteins to inhibit autophagy.

Studies have shown that oxidative stress could impair cell metabolic function to obtain abnormal liver lipid deposition, and the abnormal lipid deposition is accompanied by severe oxidative stress [8,63]. The study found that TTO levels have antioxidant properties [64]. In addition, consuming moderate levels of TTO may enhance host antioxidant capacity as a cofactor for antioxidant enzymes to increase defense against oxidative stress [23,65]. Based on these premises, we evaluated the oxidative stress status of each group. Our study found that the 100 mg/kg TTO diet enhances antioxidant enzyme activity and reduces peroxide production in the hepatopancreas compared to the control. On the contrary, 1000TTO did not improve the activity of antioxidant enzymes while it induced a significant increase in the levels of MDA and PCO, indicating that oxidative damage occurs as MDA and PCO are both marker products of oxidative damage [66]. The *Keap1*-*Nrf2* pathway has a central role in the defense against oxidative stress in vertebrates [67,68]. Studies in crustaceans have also found that oxidative stress activates the *Keap1*-*Nrf2*-mediated antioxidant pathway [69]. The activation of this pathway is mainly caused by ROS-stimulated *Nrf2* uncoupling from *Keap1* into the nucleus and binding to AREs to achieve antioxidant purposes. Our study found that the addition of 100TTO activated the *Keap1*-*Nrf2* pathway expression compared to the control group, and 1000TTO significantly increased the level of *Nrf2* expression compared to the 100TTO group. Moreover, 1000TTO enormously increased ROS production and caused mitochondrial damage accompanied by increased lipid peroxidation products (LPO and 4-HNE). This is consistent with previous findings that state that the sharp up-regulation of *Nrf2* always accompanies a vastly improved level of ROS, which may be compensatory stress [70]. Another study has shown that oxidative stress disrupted the morphology and function of mitochondria [71]. In addition, the *NF-κB* signaling pathway was found to have bidirectional regulation effects on ROS regulation [72]. *Relish* was an *NF-κB* homologous protein in invertebrate animals and was found to have oxidative stress-regulating effects [73,74]. Our results showed that 100TTO significantly increased *Relish* and its downstream factor *Imd*, whereas 1000TTO did not activate *Relish*-*Imd* expression. The above results show that 100TTO improves the antioxidant capacity by activating the *Keap1*-*Nrf2* and *Relish-Imd* signaling pathways, while the 1000TTO group only activates *Keap1*-*Nrf2* but without activating the *Relish*-*Imd* antioxidant pathway, which may also cause oxidation stress finally. Therefore, we speculated that excess TTO levels stimulate ROS production and impair the antioxidant system, thereby inducing oxidative stress that mediates excess TTO-induced mitochondrial dysfunction. Similarly, other studies have shown that high levels of essential oils cause an increase in ROS, and excess ROS induce oxidative stress damage [75,76].

The above results suggest that different TTO levels simultaneously affected oxidative stress status, autophagy levels, and the lipid metabolism process. However, the concrete underlying molecular link among them is unclear. In view of this, Pearson correlation analysis was performed to explore the correlation between the expression levels of the two oxidative stress regulatory pathways *Keap1*-*Nrf2* and *Relish*-*Imd*, lipid metabolism regulatory factors, and autophagy indicators. Through Pearson correlation analysis, two key node factors are determined; one is *Nrf2*, and the other is *Relish*.

Firstly, we found that *Nrf2* and *Pparγ* were significantly positively correlated. Martinez-Lopez et al. [77] pointed out that *Nrf2* translocates to the nucleus and activates many genes via ARE transport following oxidative stress. Oxidative stress promoted the binding of *Nrf2* to the *Pparγ* promoter and increased *Pparγ*-mediated adipogenesis [78]. At the same time, *Nrf2* was also positively correlated with *Fasn* and *Srebp1* in this study; this is consistent with previous conclusions that the activation of *Nrf2* under oxidative stress promotes adipogenesis by activating *Srebp1*, and the increase in *Fasn* expression directly causes the increase in fat synthesis [79,80]. In addition, *Nrf2* was negatively correlated with lipolysis genes *Cpt1*, *Hsl*, and *Ampkα* in this study. Past studies have found that the inhibition of lipolysis promotes adipocyte lipid accumulation [81]. Therefore, it is demonstrated that the high expression of *Nrf2* leads to increased lipid synthesis. In terms of autophagy, *Nrf2* was found to be positively correlated with autophagy-related genes, including autophagy marker *p62* and upstream regulatory genes (*Akt* and *mtor*), and negatively correlated with *Beclin1* and *Lc3*. *Nrf2* hyperactivation leads to the accumulation of *p62* and the inhibition of autophagy [82]. *Nrf2* positively regulates *Akt* expression under oxidative stress [83], and the increase in *Akt* activates the *Akt*-*mtor* pathway expression [84]. The increase in *Akt* expression will induce the dephosphorylation of *Lc3* [85]. The activation of mtor inhibits autophagy and reduces marker *Beclin1* expression [62].

The above results demonstrate that *Nrf2* is an important node between oxidative stress, autophagy, and lipid metabolism, and affects lipid deposition in the hepatopancreas simultaneously through antioxidant and lipid droplet autophagy pathways. This also explains why the 1000TTO group caused liver cell damage and excessive lipid deposition, accompanied by the high expression of *Nrf2* and inhibited autophagy and lipolysis. In addition, the down-regulation of hepatic lipid deposition in the 100TTO group may be regulated by other factors.

In terms of the other key node factor, *Relish*, correlation analysis data found that *Relish* was positively correlated with key lipolytic genes *Atgl* and *Cpt1*. Although few studies have shown that *Relish* enhances lipolysis, one study found that *Relish* was inhibited when the body fat increased in shrimp [35]. In addition, *Relish* was positively correlated with *Ulk1*, *Beclin1*, and *Lc3*, and negatively correlated with *p62*. Research has shown that *Relish* increases the activation of *atg1*(*Ulk1*) and the level of autophagy, and an increase in marker *Lc3* inevitably accompanies the increased expression of *Relish* [86]. When autophagy is blocked, the *p62* protein accumulates in the cytoplasm, so the negative correlation between *Relish* and *p62* in this study means the activation of autophagy [87].

The Ampk-mtor pathway is critical for regulating autophagy [88]. *Imd* was found negatively correlated with *mtor*, and *Relish* was positively correlated with *Ampkα*. Studies have shown that *Imd* inhibits the expression of *mTOR* [89], and mtor acts as an upstream pathway of autophagy; its inhibition will activate autophagy. Moreover, *Ampkα* acts as an essential gene for lipolysis [90]. Therefore, the *Relish*-*Imd* signal may enhance lipolytic ability by regulating *Ampk*-*mtor* to stimulate autophagy activation. *Pparγ* as an essential fat synthesis gene was negatively correlated with *Relish* in this study. A similar study also found that increased expression of *Relish* downregulates lipid synthesis through the regulation of *Pparγ* [91]. Therefore, the 100TTO group may activate autophagy by increasing the expression of *Relish* to regulate the autophagy initiation factor and down-regulating *Pparγ* to promote lipolysis.

## 5. Conclusions

In conclusion, this study demonstrates that 100 mg/kg TTO enhances antioxidant activity and induces autophagy through the *Relish*-*Imd* pathway to enhance lipid droplet breakdown and maintain lipid homeostasis. However, 1000 mg/kg TTO treatment will overexpress *Nrf2*, thus inducing the transcriptional activation of *Pparγ* and other genes to promote fat synthesis and inhibit autophagy, eventually causing lipid peroxidation and oxidative stress. This putative mechanism that can explain our results is shown in Figure 9. Our study supplies a new perspective focus on *Nrf2*, *Relish*, and the “antioxidant-autophagy-lipid metabolism” axis to reveal the regulation mechanisms of lipid metabolism by tea tree oil. Next, we will design the agonists and inhibitors of *Nrf2* and *Relish* and verify the precise mechanism of TTO to regulate the lipid metabolism of crustaceans based on the “antioxidant-autophagy-lipid-metabolism” axis at the tissue or cell level in vitro.

## Figures and Tables

**Figure 1 antioxidants-11-02260-f001:**
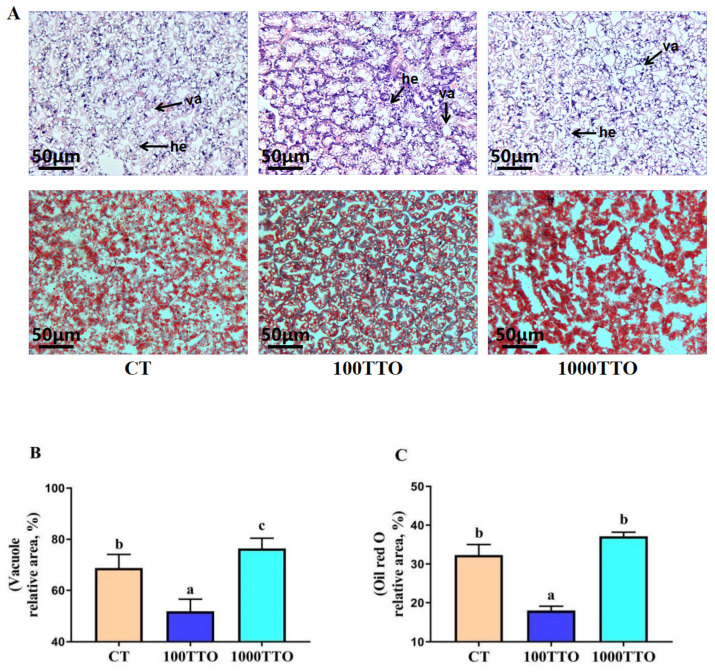
Effect of dietary TTO levels on morphology and lipid deposition in the hepatopancreas of *M. rosenbergii*. (**A**) The representative micrographs of H&E and Oil Red O-stainings (200× magnification). Va, vacuole; He, Hepatopancreas cell. (**B**) The relative vacuole area of H&E staining and (**C**) Oil Red O-staining lipid droplets. The significant difference between the three TTO levels is characterized by different lowercase letters (a, b, c) (*p* < 0.05).

**Figure 2 antioxidants-11-02260-f002:**
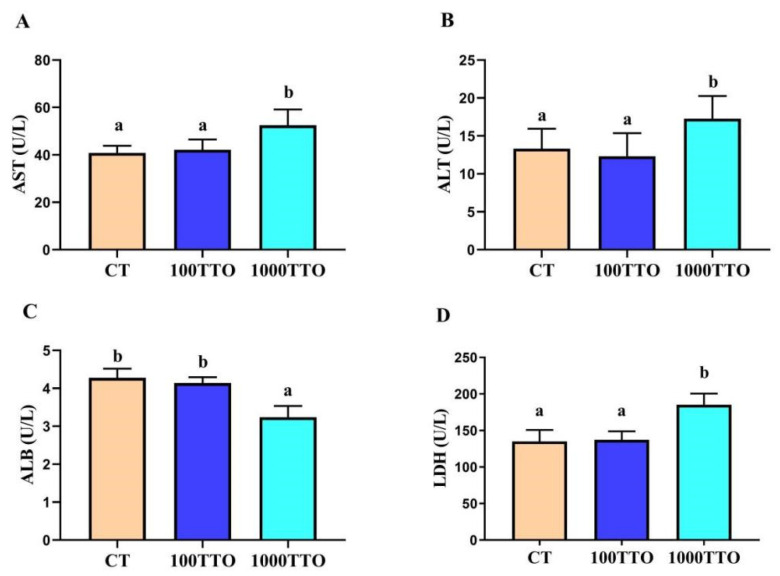
Effect of dietary TTO levels on hepatopancreatic function indicators of *M. rosenbergii*. (**A**) AST activity; (**B**) ALT activity (**C**) ALB activity; (**D**) LDH activity. The significant difference between the three TTO levels is characterized by different lowercase letters (a, b) (*p* < 0.05).

**Figure 3 antioxidants-11-02260-f003:**
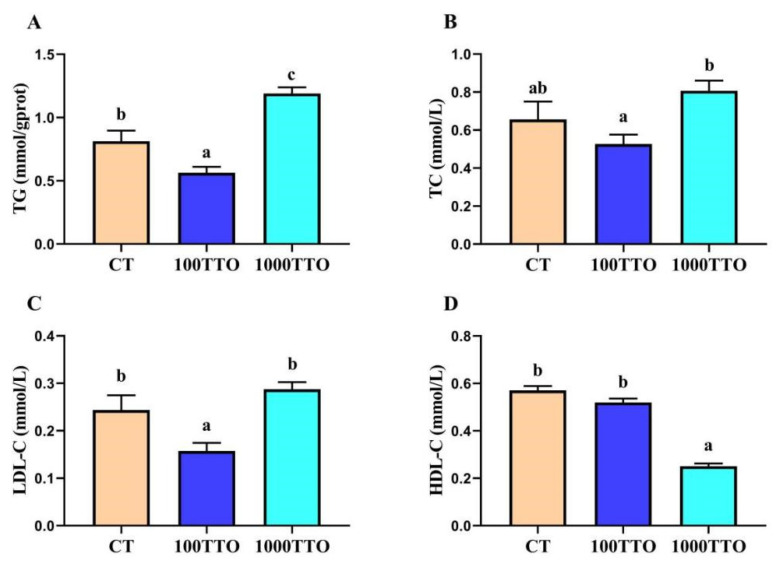
Effect of dietary TTO levels on lipid contents of *M. rosenbergii*. (**A**) TG content in hepatopancreas. (**B**–**D**) TC, LDL-C, HDL-C content in hemolymph. The significant difference between the three TTO levels is characterized by different lowercase letters (a, b, c) (*p* < 0.05).

**Figure 4 antioxidants-11-02260-f004:**
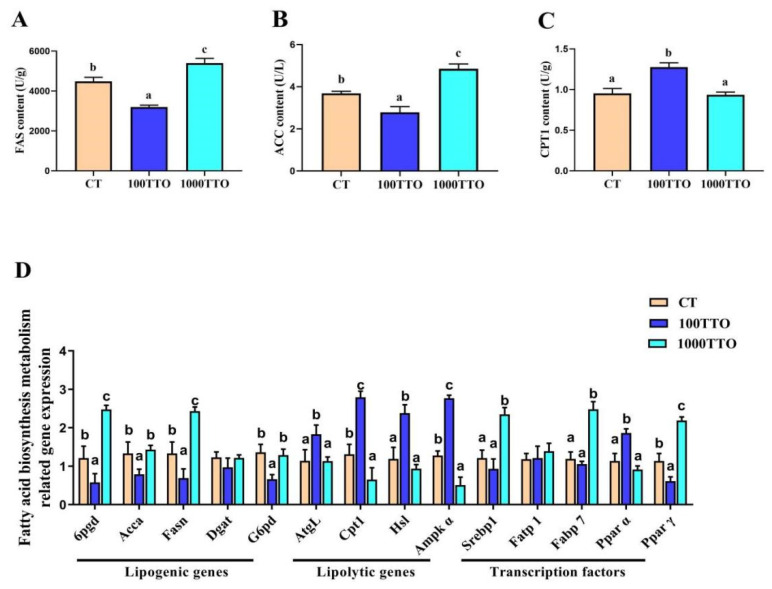
Effects of dietary TTO levels on hepatopancreatic lipid metabolism of M. rosenbergii. (**A**) FAS content; (**B**) ACC content; (**C**) CPT1 content. (**D**) mRNA expression of lipid metabolism-related genes. The significant difference between the three TTO levels is characterized by different lowercase letters (a, b, c) (*p* < 0.05).

**Figure 5 antioxidants-11-02260-f005:**
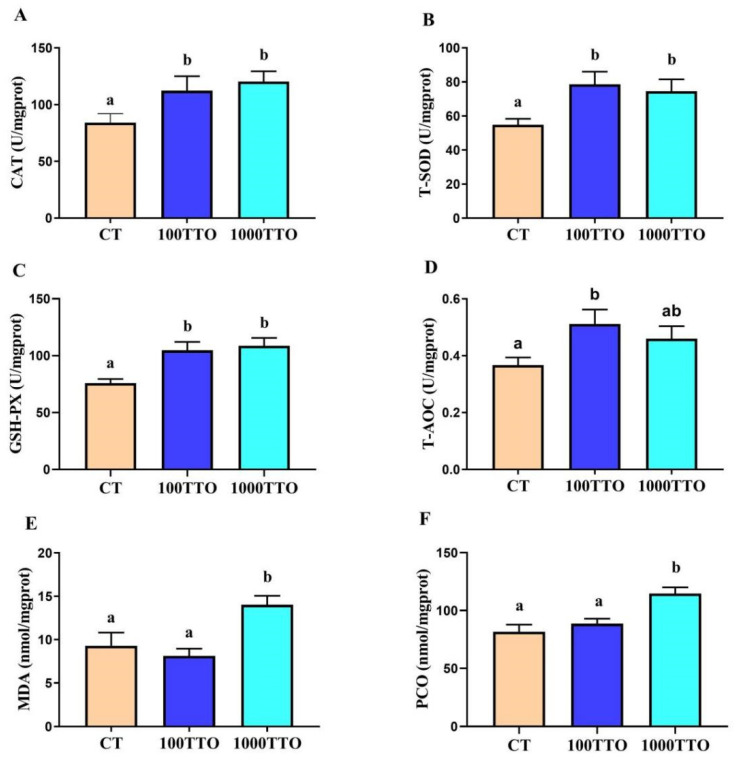
Effects of dietary TTO levels on the antioxidant level in the hepatopancreas of *M. rosenbergii*. (**A**) CAT activity; (**B**) T-SOD activity; (**C**) GSH-PX activity; (**D**) T-AOC activity; (**E**) MDA content; (**F**) PCO content. The significant difference between the three TTO levels is characterized by different lowercase letters (a, b) (*p* < 0.05).

**Figure 6 antioxidants-11-02260-f006:**
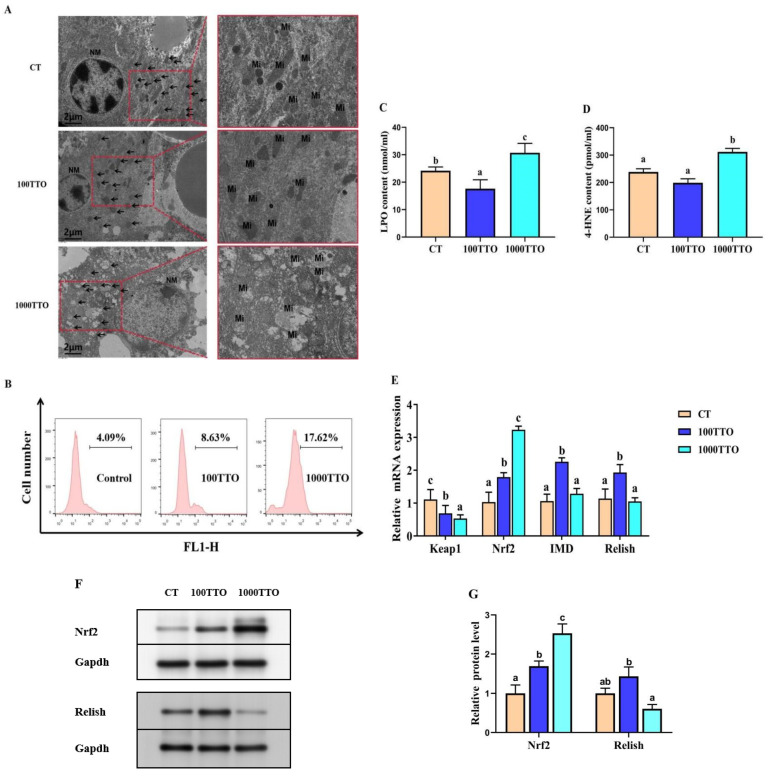
Effects of dietary TTO levels on hepatopancreatic oxidative stress of *M. rosenbergii*. (**A**) Characteristic TEM image. Arrow points to mitochondria; NM, nuclear membranes; Mi, mitochondria. (**B**) Fluorescence intensity of ROS in hemolymph displayed by flow cytometry. (**C**) LPO content; (**D**) 4-HNE content. (**E**) mRNA levels of *Keap1*, *Nrf2*, *Imd* and *Relish*. (**F**) Nrf2 and Relish protein bands under Western blot analysis. (**G**) Relative quantification of Nrf2 and Relish protein expression. The significant difference between the three TTO levels is characterized by different lowercase letters (a, b, c) (*p* < 0.05).

**Figure 7 antioxidants-11-02260-f007:**
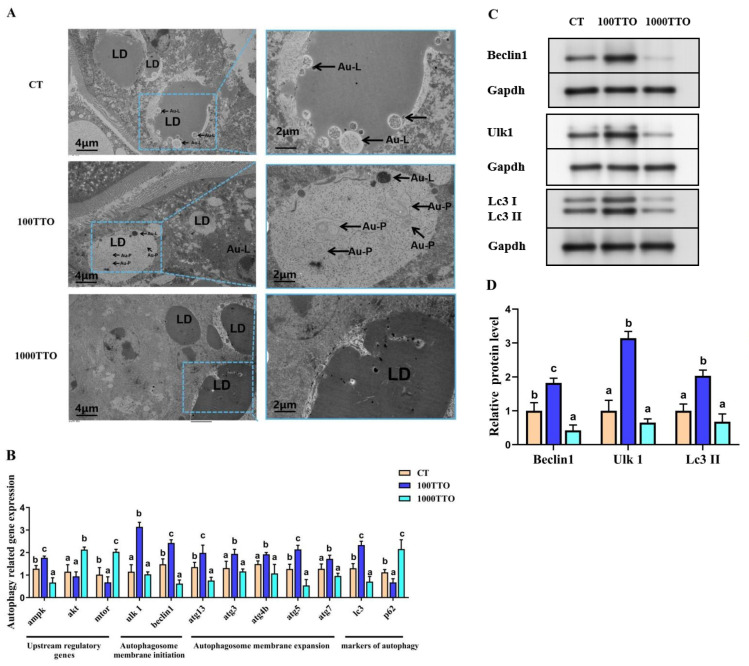
Effects of dietary TTO levels on autophagy formation in the hepatopancreas of *M. rosenbergii*. (**A**) Characteristic TEM image. Arrow points to autophagy. LD, lipid droplet; Au-P, autophagosome; Au-L, autolysosome. (**B**) mRNA levels of key autophagy-related genes. (**C**) Beclin1, Ulk1, and Lc3 protein bands under Western blot analysis. (**D**) Relative quantification of Beclin1, Ulk1, and Lc3 II protein expression. The significant difference between the three TTO levels is characterized by different lowercase letters (a, b, c) (*p* < 0.05).

**Figure 8 antioxidants-11-02260-f008:**
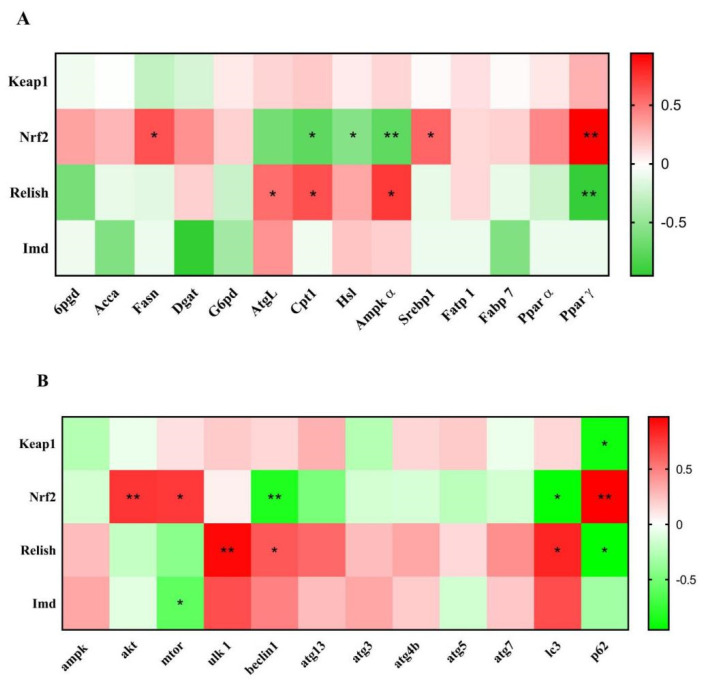
Correlation between oxidative stress with (**A**) lipid metabolism and (**B**) autophagy in mRNA levels. Note: Red represents positive correlation and green represents negative correlation. The stronger the correlation, the more intense the color. * *p* < 0.05, ** *p* < 0.01.

**Figure 9 antioxidants-11-02260-f009:**
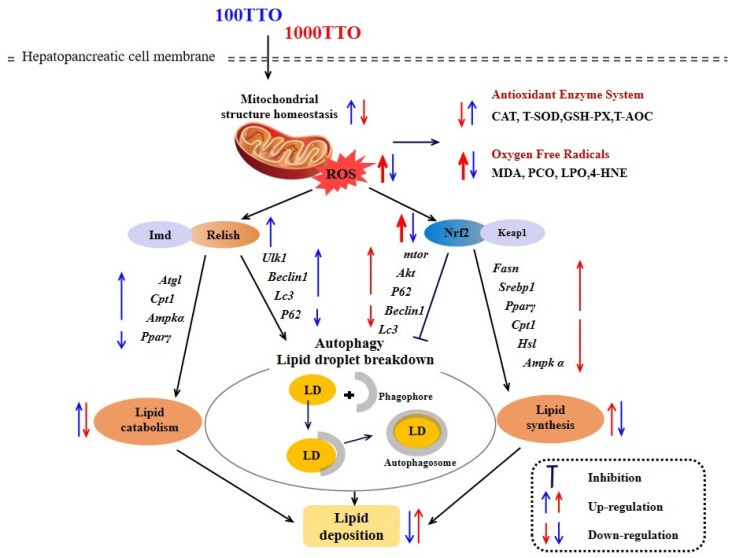
The mechanism of 100TTO and 1000TTO regulates lipid metabolism based on antioxidant factors *Relish* and *Nrf2*-autophagy axis. The blue arrows indicate the regulation effects of 100TTO; the red arrows indicate the regulation effects of 1000TTO. Arrows up indicate significant up-regulation or improvement, and arrows down indicate significant down-regulation or inhibition. Thick arrows indicate extremely significant regulation.

**Table 1 antioxidants-11-02260-t001:** The formulation and proximate composition of the experimental diet.

Ingredients (g kg^−1^)	CT	100TTO	1000TTO
Fish meal ^1^	750.00	750.00	750.00
Chellocken meal ^1^	150.00	150.00	150.00
Blood globulin powder ^1^	60.00	60.00	60.00
Shrimp meal ^1^	240.00	240.00	240.00
Soybean meal ^2^	570.00	570.00	570
Repeseed meal ^2^	300.00	300.00	300
Shrimp meal ^1^	240.00	240.00	240.00
Squid extract ^1^	90.00	90.00	90.00
Soybean oil ^2^	60.00	60.00	60.00
Fish oil ^2^	60.00	60.00	60.00
α-starch ^1^	600.00	600.00	600.00
Soy lecithin oil ^1^	30.00	30.00	30.00
Ecdysone ^1^	0.30	0.30	0.30
MCP	60.00	60.00	60.00
Premix ^3^	30.00	30.00	30.00
Choline chloride ^4^	30.00	30.00	30.00
Bentonite ^4^	29.70	29.40	26.70
10% Tea Tree Oil	0.00	3.00	30.00
Total	3000.00	3005.40	3054.00
Proximate analysis (%)			
Moisture	10.40	9.48	9.68
Crude protein	40.91	39.46	40.29
Crude lipid	9.76	9.06	9.54
Ash	17.43	16.52	16.87

Notes: ^1^ Obtained from Jiangsu Fuyuda Food Products Co., Ltd., Yangzhou, China; ^2^ Obtained from Hulunbeier Sanyuan Milk Co., Ltd., Inner Mongolia, China; ^3^ Premix supplied the following minerals (g kg^−1^) and vitamins (IU or mg kg^−1^): Pantothenate, 1000 mg; Folic acid, 165 mg; Vitamin A, 900,000 IU; Vitamin D, 200,000 IU; Vitamin E, 4500 mg; Vitamin K_3_, 220 mg; Vitamin B_1_, 320 mg; Vitamin B_2_, 1090 mg; Vitamin B_5_, 2000 mg; Vitamin B_6_, 500 mg; Vitamin B_12_, 1.6 mg; Vitamin C, 5000 mg; CuSO_4_·5H_2_O, 2.0 g; FeSO_4_·7H_2_O, 25 g; ZnSO_4_·7H_2_O, 22 g; MnSO_4_·4H_2_O, 7 g; Na_2_SeO_3_, 0.04 g; KI, 0.026 g; CoCl_2_·6H_2_O, 0.1 g. Wuxi Hanove Animal Health Products Co., Ltd. provides mineral and Vitamin additives.^4^ Obtained from Freshwater Fisheries Research Center, Chinese Academy of Fishery Sciences (CAFS).

**Table 2 antioxidants-11-02260-t002:** Effects of dietary TTO levels on growth evaluation of *M. rosenbergii*.

Index	Groups		
	CT	100TTO	1000TTO
Survival rate (%)	82.50 ^b^ ± 3.23	85.00 ^b^ ± 5.50	72.50 ^a^ ± 4.62
Initial weight (g)	0.14 ± 0.01	0.15 ± 0.01	0.15 ± 0.00
Final weight (g)	2.84 ± 0.10 ^a^	3.05 ± 0.14 ^ab^	3.44 ± 0.44 ^b^
Weight gain rate (%)	1839.82 ± 51.39 ^a^	1942.06 ± 117.08 ^ab^	2130.23 ± 107.98 ^b^
Specific growth rate (%/day)	6.18 ± 0.05 ^a^	6.27 ± 0.12 ^a^	6.42 ± 0.26 ^b^
Feed conversion ratio	1.33 ± 0.04 ^b^	1.15 ± 0.01 ^a^	1.26 ± 0.04 ^b^
Hepatopancreas index (%)	7.23 ± 1.06 ^b^	5.73 ± 0.86 ^a^	10.73 ± 0.42 ^c^

Note: Data are expressed as means with SEM. Means in the same line with different superscripts (a, b and c) are significantly different (*p* < 0.05).

## Data Availability

Data are contained within the article.

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
