# Peer review of "Tea Tree Oil Mediates Antioxidant Factors Relish and Nrf2-Autophagy Axis Regulating the Lipid Metabolism of Macrobrachium rosenbergii"

_antioxidants, 2022, doi:10.3390/antiox11112260_

Round 1

Reviewer 1 Report

The submitted paper titled “Tea Tree Oil mediates antioxidant factors Relish and Nrf2 - autophagy axis regulating lipid metabolism of Macrobrachium rosenbergii” presents a thorough investigation of the effects of supplementation of feed with Tea Tree Oil (TTO) on hepatopancreatic characteristics related to lipid metabolism in M. rosenbergii prawns. The paper represents an extensive study and provides interesting, clear and well documented conclusions regarding the effects of TTO on the antioxidant-autophagy axis. I recommend the publication of the submitted paper with only minor comments.

-       The description provided for the assays of enzymes, and some lipids and metabolites includes only the parameter assayed and the manufacturer of the kit, stating at the end of the paragraphs "Specific methods according to the manufacturer's instructions". I consider this information insufficient, and more information should be provided stating at least the respective substrates and the principles of the methods, either in the Materials and Methods, or in the Supplementary Section.

-       Scientific nomenclature of all living organisms should be in Italics, something the authors have missed in a few occasions in the MS (Melaleuca alternifolia, Calocedrus formosana, M. rosenbergii)

-       Starting a sentence with a number should be avoided in a couple of cases in the Materials and Methods: “360 freshwater prawns”, “3 prawns per replicate”

-       In Page 5, Line 228, “Data analysis” is presented and not the “Date analysis”, as it is written in the MS

-       Page 2, Line 64, “ROS induce” instead of “ROS induces”

-       Page 2, Line 78, the sentence is unclear, please rephrase

Author Response

Reply to Reviewer 1:

The submitted paper titled “Tea Tree Oil mediates antioxidant factors Relish and Nrf2 - autophagy axis regulating lipid metabolism of Macrobrachium rosenbergii” presents a thorough investigation of the effects of supplementation of feed with Tea Tree Oil (TTO) on hepatopancreatic characteristics related to lipid metabolism in M. rosenbergii prawns. The paper represents an extensive study and provides interesting, clear and well documented conclusions regarding the effects of TTO on the antioxidant-autophagy axis. I recommend the publication of the submitted paper with only minor comments.

Response: Thanks to the reviewers for their recognition and careful review of my manuscript and making many meaningful points, I have revised the article based on your comments. The following are my responses to the comments:

  1. The description provided for the assays of enzymes, and some lipids and metabolites includes only the parameter assayed and the manufacturer of the kit, stating at the end of the paragraphs "Specific methods according to the manufacturer's instructions". I consider this information insufficient, and more information should be provided stating at least the respective substrates and the principles of the methods, either in the Materials and Methods, or in the Supplementary

Response 1: Thank you for your meaningful and valuable suggestions. We have supplemented these sections as follows:

Catalase (CAT), total superoxide dismutase (T-SOD), glutathione peroxidase (GSH-Px), total antioxidant capacity (T-AOC) and lactate dehydrogenase (LDH) activities, protein carbonyl (OH) and triglyceride (TG) content were analyzed by using the kits from Nanjing Jiancheng Institute of Bioengineering (Nanjing, China). The general method is to suspend hepatopancreas in 50 mM potassium phosphate buffer with 0.5 mM EDTA at pH of about 7.0. Homogenates of the hepatopancreas tissues were performed in microcentrifuge tubes using a plastic hand pestle for 5 min. At the termination of the homogenization process, allinclusive samples were subjected to centrifuge set at 2000g for 15 min at 2°C. The supernatant resultant was removed for use in enzyme analyses. The all-inclusive enzyme activity analyses were conducted using a Spectra Max Plus spectrophotometer (Molecular Devices, Menlo Park, CA, USA). Malonaldehyde (MDA) content in the hepatopancreas was estimated using the thiobarbituric acid (TBA) test (Satoh, 1978) by the assay kit from Nanjing Jiancheng Bioengineering Institute, China. MDA assay method is the same as enzyme. All specific experimental methods according to the manufacturer's instructions.

Content of several lipid metabolism-related enzymes (fatty acid synthase ,FAS; Acetyl-CoA carboxylase,ACC; carnitine palmitoyltransferase 1,CPT 1) and peroxide (Lipid peroxidation,LPO; 4-hydroxynonenal,4-HNE) were determined by ELISA kits specific for freshwater shrimp (Shanghai mlbio Biotechnology Co., Ltd., China) according to the manufacturer's instructions. Briefly, the hepatopancreas supernatant was mixed with the reactants and the absorbance corrected at 430 nm and 650 nm was recorded. The Spectra Max Plus spectrophotometer was used for all of the Elisa’s technical readings of optical density.

  1. Scientific nomenclature of all living organisms should be in Italics, something the authors have missed in a few occasions in the MS (Melaleuca alternifolia,Calocedrus formosana,  rosenbergii)

Response 2: Thanks for your careful review. We have made changes in the manuscript based on your comments. We used italics for the scientific nomenclature of all living organisms in the manuscript.

  1. Starting a sentence with a number should be avoided in a couple of cases in the Materials and Methods: “360 freshwater prawns”, “3 prawns per replicate”

Response 3: Thanks for your meaningful suggestion. We rewrote these sentences. As the following sentence:

“Put 360 freshwater prawns (initial body weight: 0.39 ± 0.01 g) in 9 vats (R × H, 1.0 m × 1.5 m), 40 prawns per vat.”

“The hepatopancreas of 3 prawns in each replicate was anatomized for samples.”

  1. In Page 5, Line 228, “Data analysis” is presented and not the “Date analysis”, as it is written in the MS

Response 4: Thanks for your careful review. We changed “Date analysis” to “Data analysis”.

  1. Page 2, Line 64, “ROS induce” instead of “ROS induces”

Response 5: Thank you for your suggestion and careful review. We changed “ROS induces” to “ROS induce”.

  1. Page 2, Line 78, the sentence is unclear, please rephrase

Response 6: Thank you for your suggestion. We rewrite the sentence as “Additionally, study found that an excessive increase in ROS is a critical factor in lipid excess in the liver.”

We have highlighted changes to the manuscript in yellow, including modifications based on comments from other reviewers.

Reviewer 2 Report

This article presents the results of a study by Liu et al. that aimed to evaluate the effects of tea tree oil on the regulation of lipid metabolism of Macrobrachium rosenbergii through the “antioxidant-autophagy-lipid metabolism” axis. This topic is interesting, and the work carried out is well presented.

Below you can find some questions to take into consideration, which are presented to contribute to the improvement of the submitted version of article.

Please revise the text because Latin names of the species should be presented in italic, and it is not always the case. In addition, the text presents several inconsistencies in the verbal tense used (see below an example, line 178). You should avoid using both present and past tense when describing the methods and avoid referring to “I” or “We”. There are also some inconsistencies in the use of uppercase/lowercase, and italicized designations that should be reviewed.

Several parameters that were analyzed had already been evaluated in previous work published by the authors; however, in some cases, the previous work shows that results were significantly different (e.g., AST, ALT, TC, MDA) while in this study they aren’t. How do you explain that? I noticed that the diet was different. Was there any reason for changing diet composition? Could this be an explanation?

Line 78: “Studies toxicologically, an increase in ROS is a critical factor…” What do you mean by “Studies toxicologically,…”? Please revise the sentence

Line 105: “feeding were presented…” should be “feeding are presented…”

Lines 153-154: should be “…and 2.5% glutaraldehyde solution, respectively for histochemical and ultrastructural analysis.”

Lines 161-162: “Content of several lipid metabolism-related enzymes, including Lipid peroxidation 161 (LPO), 4-hydroxynonenal (4-HNE), fatty acid synthase (FAS),…” please revise as Lipid peroxidation 161 (LPO), 4-hydroxynonenal (4-HNE) are not enzymes

Line 167: ”total antioxidant capacity (T-AOC)” is not an enzyme either; please revise both paragraphs (lines 161-170) to separate enzymes from the other parameters that were determined

Line 175: “1×106/mL.” should be “1×106/mL.”

Line 178: “Next, wash the cells three times using…” should be “Next, cells were washed three times using…”

Figure 1, panel A: the meaning of the letters/arrows should be presented in the legend.

Results are presented in a very brief way, without any reference to their meaning, which could be useful for readers with less experience in this topic.

Did you evaluate the levels of the enzymes or their activity? And why? What do they mean in terms of hepatopancreatic cell function?

What about the “key genes” mentioned in the legend pf the figure 6? What is their significance in relation to the oxidative stress?

Discussion: You have several incomplete and confusing sentences. Find below some examples. This section needs careful and in-depth review.

Lines 408-409: Did you check your hypothesis, for example by evaluating the muscles’ growth? If not, do you intend do it in future experiments? How?

Lines 412-413: How is it possible to consider that the overall growth performance improvement is only due to the increase in hepatopancreas weight? Is there any evidence that support this?

Lines 424-425: “…the appropriate TTO levels could reverse this phenomenon.” Did you observe this? What are the “appropriate TTO levels”?

Line 433: “…less vacuolation, and clearer nuclear.” Do you mean “clearer nucleus”?

Line 444: “…the 1000TTO group may destroy hepatopancreatic function…” Please check the sentence. It is not the group that destroys the hepatopancreatic function but the treatment with the dosage used in this group.

Lines 446-447: “…As the significantly elevated AST, ALT, and LDH are markers of impaired hepatopancreatic function…” Please rewrite or complete the sentence. Then, the following sentence (about ALB) is also incomplete. You do not discuss the AST and ALT results.

Lines 453-457: you are discussing results related to FAS and ACC and then you write “…and increased in the 1000TTO group CPT1 is a rate-limiting enzyme in fatty acid β-oxidation”. Confusing. Please check punctuation and rewrite so that your message is clear.

Conclusions: Check the verbal tenses (past, present, or future? Not always correct); Interesting mechanism of action proposal (figure 9).

Author Response

Reply to the Reviewer 2:

  1. Please revise the text because Latin names of the species should be presented in italic, and it is not always the case. In addition, the text presents several inconsistencies in the verbal tense used (see below an example, line 178). You should avoid using both present and past tense when describing the methods and avoid referring to “I” or “We”. There are also some inconsistencies in the use of uppercase/lowercase, and italicized designations that should be reviewed.

Response 1: Thank you for your meaningful and valuable suggestions. We have revised the manuscript according to your requirements. We have unified tenses, removed the use of "I"&"WE", and checked the gene case and italics in the manuscript. The revisions are provided in our resubmitted manuscript.

  1. Several parameters that were analyzed had already been evaluated in previous work published by the authors; however, in some cases, the previous work shows that results were significantly different (e.g., AST, ALT, TC, MDA) while in this study they aren’t. How do you explain that? I noticed that the diet was different. Was there any reason for changing diet composition? Could this be an explanation?

Response 2: Thank you for your suggestion. We speculate that the reasons include the following aspects: 1. The mean initial weight (originally, 0.39 ± 0.01 g; Now, 0.15 ± 0.01 g) and mean final weight of prawns in the two feeding experiments were different , in addition to the weight gain rate is not the same (originally, 4165.31%; Now, 1942.06%), indicating that the growth stage, the physical condition or health status of prawns during the whole experiment was different, so their physiological indicators may have different levels of response to external interventions. 2. The two experiments' nutritional and raw material compositions are different. The original formula is 40% proteins and 10% lipids, and this formula is 40% proteins and 9% lipids, which may also lead to the difference of nutritional status and overall health status of shrimp. 3. Although there was no significant difference in these four indexes (such as AST, ALT, TC, MDA), the effects of 100mg/kgTTO supplementation were consistent, that is, they all improved the antioxidant capacity and liver health degree of shrimp. The feed formula was changed because the two breeding experiments were carried out in 2019 and 2020, respectively. Due to the limitation of the types of experimental raw materials in those years, we fine-tuned the formula composition.

  1. Line 78: “Studies toxicologically, an increase in ROS is a critical factor…” What do you mean by “Studies toxicologically,…”? Please revise the sentence

Response 3: Thank you for your suggestion. We rewrite the sentence as “Additionally, study found that an excessive increase in ROS is a critical factor in lipid excess in the liver.

  1. Line 105: “feeding were presented…” should be “feeding are presented…”

Response 4: Thanks for your careful review. We changed “feeding were presented” to “feeding are presented”.

  1. Lines 153-154: should be “…and 2.5% glutaraldehyde solution, respectively for histochemical and ultrastructural analysis.”

Response 5: Thanks for your careful review. We changed the sentence as “…and 2.5% glutaraldehyde solution, respectively for histochemical and ultrastructural analysis.”.

  1. Lines 161-162: “Content of several lipid metabolism-related enzymes, including Lipid peroxidation 161 (LPO), 4-hydroxynonenal (4-HNE), fatty acid synthase (FAS),…” please revise as Lipid peroxidation 161 (LPO), 4-hydroxynonenal (4-HNE) are not enzymes. ; Line 167: ”total antioxidant capacity (T-AOC)” is not an enzyme either; please revise both paragraphs (lines 161-170) to separate enzymes from the other parameters that were determined

Response 6: Thanks for your meaningful suggestion. We have modified the sentence. As follows:

Content of several lipid metabolism-related enzymes (fatty acid synthase ,FAS; Acetyl-CoA carboxylase,ACC; carnitine palmitoyltransferase 1,CPT 1) and peroxide (Lipid peroxidation,LPO; 4-hydroxynonenal,4-HNE) were determined by ELISA kits specific for freshwater shrimp (Shanghai mlbio Biotechnology Co., Ltd., China) according to the manufacturer's instructions. We also change the section title to " Antioxidant, peroxide and lipid parameters analysis".

  1. Line 175: “1×106/mL.” should be “1×106/mL.”

Response 7: Thanks for your careful review. We changed “1×106/mL.” to “1×106/mL.”.

  1. Line 178: “Next, wash the cells three times using…” should be “Next, cells were washed three times using…”

Response 8: Thanks for your meaningful suggestion. We changed "Next, wash the cells three times using…" to "Next, cells were washed three times using…".

  1. Figure 1, panel A: the meaning of the letters/arrows should be presented in the legend.

Response 9: Thanks for your suggestion. We added the meaning of the abbreviation in the legend of Figure 1, see in Figure 1.

  1. Resultsare presented in a very brief way, without any reference to their meaning, which could be useful for readers with less experience in this topic. Did you evaluate the levels of the enzymes or their activity? And why? What do they mean in terms of hepatopancreatic cell function? 

Response 10: Thanks for your meaningful suggestion. We changed the sentence to “Compared to control, 1000TTO induced the significantly increased activity levels of hepatopancreatic function related enzymes (AST, ALT and LDH), and the significantly decreased ALB level (P < 0.05).”. In addition, in lines 484-488 of the discussion section, we supplemented the significance of these enzyme activities changed for hepatocyte function and described the meanings of these indicators,as follow:

The significantly elevated AST, ALT, and LDH in hemolymph are markers of impaired hepatopancreatic function. When the hepatocytes are damaged, AST and ALT will enter the blood from the hepatocytes and cause their content to increase (Dai et al., 2008; Gao et al., 2021). ALB is mainly synthesized in the hepatopancreas and is an important immune protein. Hepatopancreatic function damage will bring about a significant reduction in ALB levels. (Vincent et al., 2021). Combined with AST, ALT, LDH and ALB results, it was proved that 1000TTO causes hepatopancreas dysfunction. 

  1. What about the “key genes” mentioned in the legend pf the figure 6? What is their significance in relation to the oxidative stress?

Response 11: Thanks for your suggestion. We changed “key genes” to “Keap1, Nrf2, Imd and Relish” in the legend pf the figure 6. These four genes are the genes related to oxidative stress of Macrobrachium rosenbergii in our experiment, their gene expression levels regulate oxidative stress.

  1. Discussion: You have several incomplete and confusing sentences. Find below some examples. This section needs careful and in-depth review.Lines 408-409: Did you check your hypothesis, for example by evaluating the muscles’ growth? If not, do you intend do it in future experiments? How?

Response 12: Thanks for your careful reviewing. We actually measured the ratio of muscle to total weight in each group, but it's not in the article. Because we have carried out studies on the regulation of tea tree essential oil on muscle quality (Protein synthesis, muscle fiber development, flavor substance synthesis; Unpublished). We have revised this sentence, as follows:

“So, combined with our undisclosed results about muscles, we speculated that the 100 mg/kg TTO diet may promote the other tissues’ growth (e.g. muscles) instead of hepatopancreas.”

  1. Lines 412-413: How is it possible to consider that the overall growth performance improvement is only due to the increase in hepatopancreas weight? Is there any evidence that support this?

Response13: Thanks for your suggestion. When calculating HSI, the hepatopancreas of each group were weighed separately, and it was found that the hepatopancreas of 1000TTO group were significantly heavier than those of the other two groups. Moreover, as mentioned in the previous answer, 1000TTO group had less muscle gain than 100TTO group, so we speculated that 1000TTO could improve growth by increasing hepatopancreas weight.

  1. Lines 424-425: “…the appropriate TTO levels could reverse this phenomenon.” Did you observe this? What are the “appropriate TTO levels”?

Response 14: Thanks for your suggestion. We change “appropriate TTO levels” to “100 mg/kg TTO”, and change “excessive TTO levels” to “1000 mg/kg TTO”.

15 Line 433: “…less vacuolation, and clearer nuclear.” Do you mean “clearer nucleus”?

Response 15: Thanks for your careful reviewing. We have changed “clearer nuclear” to “clearer nucleus”.

16 Line 444: “…the 1000TTO group may destroy hepatopancreatic function…” Please check the sentence. It is not the group that destroys the hepatopancreatic function but the treatment with the dosage used in this group.

Response 16: Thanks for your careful reviewing. We have changed “the 1000TTO group” to “the 1000TTO treatment”.

17 Lines 446-447: “…As the significantly elevated AST, ALT, and LDH are markers of impaired hepatopancreatic function…” Please rewrite or complete the sentence. Then, the following sentence (about ALB) is also incomplete. You do not discuss the AST and ALT results.

Response 17: Thanks for your meaningful suggestion. We rewrote the paragraph as follows:

The significantly elevated AST, ALT, and LDH are markers of impaired hepatopancreatic function. When the hepatocytes are damaged, AST and ALT will enter the blood from the hepatocytes and cause their content to increase (Dai et al., 2008; Gao et al., 2021). ALB is mainly synthesized in the hepatopancreas and is an important immune protein. Hepatopancreatic function damage will bring about a significant reduction in ALB levels. (Vincent et al., 2021). Combined with AST, ALT, LDH and ALB results, it was proved that 1000TTO causes hepatopancreas dysfunction.

  1. Lines 453-457: you are discussing results related to FAS and ACC and then you write “…and increased in the 1000TTO group CPT1 is a rate-limiting enzyme in fatty acid β-oxidation”. Confusing. Please check punctuation and rewrite so that your message is clear.

Response 18: Thanks for your careful reviewing. We added punctuation and connectives, as follows:

“and increased in the 1000TTO group. In addition, CPT1 is a rate-limiting enzyme in fatty acid β-oxidation”.

  1. Conclusions: Check the verbal tenses (past, present, or future? Not always correct); Interesting mechanism of action proposal (figure 9).

Response 19: Thanks for your careful reviewing. We changed the tense of this passage, as follows:

In conclusion, this study demonstrates that 100 mg/kg TTO enhances antioxidant activity and induces autophagy through the Relish-Imd pathway, to enhance lipid droplet breakdown and maintain lipid homeostasis. While 1000 mg/kg TTO treatment will overexpress Nrf2, thus inducing the transcriptional activation of Pparγ and other genes to promote fat synthesis, and inhibit autophagy, eventually causing lipid peroxidation and oxidative stress. This putative mechanism that can explain our results is shown in Figure 9. Our study supplies a new perspective focus on Nrf2 and Relish, and the "antioxidant-autophagy-lipid metabolism" axis to reveal the regulation mechanisms of lipid metabolism by tea tree oil.

We have highlighted changes to the manuscript in yellow, including modifications based on comments from other reviewers.

Reviewer 3 Report

Article

Manuscript ID: antioxidants-1974174

Tea Tree Oil mediates antioxidant factors Relish and Nrf2 - autophagy axis regulating lipid metabolism of Macrobrachium rosenbergii

Mingyang Liu, Xiaochuan Zheng, Cunxin Sun , Qunlan Zhou, Bo Liu, and Pao Xu

In this experimental study Liu et al analyzed the effects of tea tree oil on lipid metabolism of Macrobrachium rosenbergii, by evaluating the interference with antioxidant factors of autophagy axis. They observed that low doses of tea tree oil could ameliorate or did not affect the lipid metabolism while high doses altered lipid homeostasis, inducing oxidative stress and lipid accumulation.

The study analyzes the interesting topic of the role of natural compounds on lipid metabolism, and the paper is suitable for the special issue “Oxidative Stress in Aquatic Organisms”, however criticisms arise, and the manuscript should be revised before publication.

Major and minor comments

The use of the crustacean model should be better argued starting from the introduction. What is the similarity or the difference with mammalian models, i.e., rodent models? This is important considering the implications in the development of therapeutic strategies against oxidative stress, as mentioned at the end of the introduction.

In this model they observed beneficial effects at the lower doses while higher doses cause damage to hepatopancreas, thus they are toxic. Did authors evaluate intermediate doses to assess “threshold dose”?

In Figures 6F and 7C, a graph with the semiquantitative assessment of the protein expression should be added. Blots should report the molecular weight of proteins. The whole blot of relish is not present in the supplementary material.

Line 306: In general, only the name of genes requires cursive font, while the name of proteins requires normal font, and, if an acronym indicating a protein is reported, uppercase characters should be used. Please, check carefully throughout the manuscripts.

Line 356: shrimps or prawns?

The limitations of this study should be discussed. 

Author Response

Reply to the Reviewer 3:

  1. The use of the crustacean model should be better argued starting from the introduction. What is the similarity or the difference with mammalian models, i.e., rodent models? This is important considering the implications in the development of therapeutic strategies against oxidative stress, as mentioned at the end of the introduction.

Response 1: Thanks for your meaningful suggestion. We added the following paragraph at the beginning of the introduction in Lines 52-61:

“The hepatopancreas is the most important tissue for lipid synthesis, metabolism, and deposition in the crustacean. The hepatopancreas of decapods exerts pancreatic, hepatic and intestinal functions compared with mammals (Ruan et al., 2022), and it is more sensitive and intolerant to high lipid deposition than the liver of mammals (Huang et al., 2022). Excessive deposition of lipids can lead to steatosis-like lesions of the hepatopancreas (Vogt, 2021). The pathological symptoms of lipid metabolism disorders in the crustacean are similar to that of mammalian nonalcoholic fatty liver disease, manifested by abnormal blood lipid parameters and disordered expression of hepatic lipid synthesis and catabolism genes (Acca, Fasn, Pparα, Cpt1) (Ishida et al., 2021; Shi et al., 2021).”. 

  1. In this model they observed beneficial effects at the lower doses while higher doses cause damage to hepatopancreas, thus they are toxic. Did authors evaluate intermediate doses to assess “threshold dose”?

Response 2: Thanks for your careful reviewing. We previously conducted a trial to explore the effects of TTO supplementation at doses ranging from 0 to 400 mg/kg on the physiological health and nonspecific immunity of M. rosenbergii. We determined that the most appropriate supplemental dose was 100mg/kg, so 100TTO was selected as the "positive treatment group" in this study. In addition, compared with the control group, 400mg/kg TTO supplementation did not reduce the antioxidant performance of Macrobrachium rosenbergii compared with the control group in the previous report (Liu et al., 2022). Therefore, 400mg/kg TTO supplementation is not the "critical dose" of TTO supplementation. Thus, to successfully construct the model of hepatopancreas toxicity induced by high-dose tea tree oil, We directly selected 10 times the dose (1000mg/kg) of tea tree oil to add and successfully made the model.

Liu, M.Y.; Gao, Q.; Sun, C.; Liu, B.; Liu, X.; Zhou, Q.; et al. Effects of dietary tea tree oil on the growth, physiological and non-specific immunity response in the giant freshwater prawn (Macrobrachium rosenbergii) under high ammonia stress. Fish & Shellfish Immunol. 2022a, 120, 458–469.

  1. In Figures 6F and 7C, a graph with the semiquantitative assessment of the protein expression should be added.Blots should report the molecular weight of proteins. The whole blot of relish is not present in the supplementary material.

Response 3: Thanks for your careful review and advice. We added the figure of relative protein expression. See Fig. 6G and Fig. 7D. The Relish BLOT is NF-κB in the supplementary material, and the NF-κB gene is represented as Relish in invertebrate animals.

  1. Line 306: In general, only the name of genes requires cursive font, while the name of proteins requires normal font, and, if an acronym indicating a protein is reported, uppercase characters should be used. Please, check carefully throughout the manuscripts.

Response 4: Thanks for your careful reviewing and meaningful advise. We have revised the gene names in the article to cursive font and the protein names to normal font, and change the protein initials to uppercase characters.

  1. Line 356: shrimps or prawns?

Response 5: Thanks for your careful reviewing. We changed “ shrimp” to “ prawn”.

  1. The limitations of this study should be discussed.

Response 6: Thanks for your suggestion. We add the following sentence at the end of the manuscript. As follow:

“Next, we will design the agonists and inhibitors of Nrf2 and Relish and verify the precise mechanism of tea tree oil to regulate the lipid metabolism of crustaceans based on the axis of "antioxidant autophagy lipid metabolism" at the tissue or cell level in vitro.”

We have highlighted changes to the manuscript in yellow, including modifications based on comments from other reviewers.

Round 2

Reviewer 3 Report

The authors performed the requested modifications and addressed the reviewer’s comments.